# On the onset of memorization to generalization transition in diffusion models

## Abstract

As the training set size increases, diffusion models have been observed to transition from memorizing the training dataset to generalizing to the underlying data distribution. To study this phenomenon more closely, here, we first present a mathematically principled definition of this transition: the model is said to be in the generalization regime if the generated distribution is closer to the sampling distribution compared to the probability distribution associated with a Gaussian kernel approximation to the training dataset. Then, we develop an analytically tractable diffusion model that features this transition when the training data is sampled from an isotropic Gaussian distribution. Our study reveals that this transition occurs when the distance between the generated and underlying sampling distribution begins to decrease rapidly with the addition of more training samples. This is to be contrasted with an alternative scenario, where the model's memorization performance degrades, but generalization performance doesn't improve. We also provide empirical evidence indicating that realistic diffusion models exhibit the same alignment of scales.

## 1 Introduction

In recent years, generative artificial intelligence has made tremendous advancements—be it image, audio, video, or text domains—on an unprecedented scale. Diffusion models (Sohl-Dickstein et al., 2015; Song & Ermon, 2019; Ho et al., 2020) are among the most successful frameworks, serving as the foundation for prominent content generation tools such as DALL-E (Ramesh et al., 2022), Stable Diffusion (Rombach et al., 2022), Imagen (Saharia et al., 2022), Sora (OpenAI, 2024) and numerous others. However, the factors that contribute to the strong generalization capabilities of diffusion models, as well as the conditions under which they perform optimally, remain open.

In this paper, we will focus on a particular generalization behavior that diffusion models exhibit. Empirical observations show that for small number of training samples, diffusion models memorize the training data (Somepalli et al., 2022; Carlini et al., 2023). As the training dataset size increases, they transition from memorizing data to a regime where it can generate new samples from the underlying distribution (Kadkhodaie et al., 2023). This has been termed the memorization-to-generalization transition. What is the nature of this transition?

In this paper, we take steps towards better understanding of this phenomenon. Specifically, we provide a mathematically precise definition of the memorization-to-generalization transition in terms of distances between the training and generated distribution, denoted as $E_{TG}$, and the distance between the original underlying distribution and generated distribution, denoted as $E_{OG}$. We say that the diffusion model is starting to generalize if the probability of $\Delta = E_{TG} - E_{OG} > 0$ is very close to unity implying that the generated distribution exhibits greater proximity to the original underlying distribution relative to the training dataset.

On the onset of memorization to generalization transition individually $E_{TG}, E_{OG}$ might show very different behaviours as shown in Fig. 1. For example the scenario (a) indicates that the model is failing to memorize because of faster increase in $E_{TG}$ compared to $E_{OG}$ with the increase of train dataset size $n$. However in this case the model does not learn to generalize well because $E_{OG}$ remains almost constant during the transition. The scenarios in (b) and (c) present other possibilities. Empirically, we rule out the possibility (b) in a realistic diffusion model. In this paper we show that

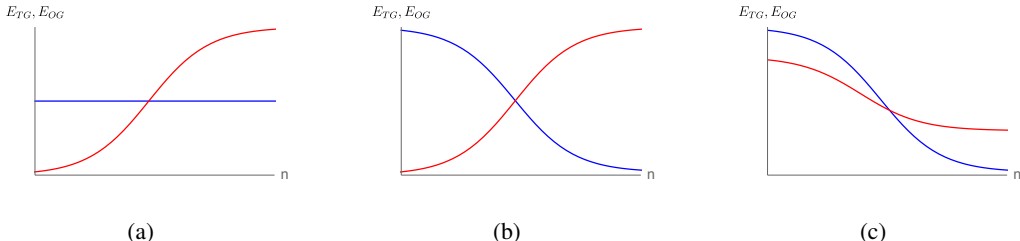

Figure 1: Hypothetical relationships between the training dataset size, denoted as $n$, and two key distributional distances is graphically represented. The distance between the training and generated probability distributions, $E_{TG}$, is depicted in red, while the distance between the original and generated probability distributions, $E_{OG}$, is illustrated in blue. Three hypothetical scenarios, labeled (a), (b), and (c), are proposed to elucidate the potential transition from memorization to generalization. The present study aims to determine which of these scenarios most accurately reflects the behavior of a realistic diffusion model.

for a diffusion model scenario (c) is the appropriate description, i.e., memorization to generalization transition happens on the onset of rapid decrease in $E_{OG}$.

## 1.1 OUR CONTRIBUTIONS

Our main contributions in this paper are as follows:

1. Given a finite number of samples from a probability distribution, the true distribution can be approximated by a $L^2$-distance optimal Gaussian kernel. We observe that in the context of higher dimensional statistics, the variance of the kernel coincides with that mixing time for the samples under Ornstein-Uhlenbeck forward diffusion process (Biroli et al., 2024).

2. Using the observation above, we formulate a new, mathematically rigorous metric, namely probability of $\Delta > 0$, for characterizing the memorization to generalization transition.

3. We present an analytically tractable diffusion model that features a transition from memorization to generalization, as measured by the aforementioned metric.

4. We show that as the size of the training dataset increases, the memorization to generalization transition in the model introduced above occurs at the same scale as the onset of a rapid fall phase in the generalization error $E_{OG}$.

5. We hypothesize that the alignment of memorization-to-generalization transition and onset of rapid convergence is a generic property of diffusion models. We test this hypothesis on a realistic U-Net based non-linear diffusion model.

Overall, our study shows that memorization-to-generalization transition in diffusion models occurs on onset of fast convergence of the generated distribution to the sampling distribution of the training data.

## 1.2 RELATED WORKS

The idea of diffusion based generative models originated in the pioneering work of Sohl-Dickstein et al. (2015). Subsequently, diffusion models are made scalable for real world image generation (Song & Ermon, 2019; Ho et al., 2020; Kadkhodaie & Simoncelli, 2020; Song et al., 2021a). The quality of the generated distribution was further enhanced through guided diffusion at the cost of reduced diversity (Dhariwal & Nichol, 2021; Ho & Salimans, 2022; Wu et al., 2024; Bradley & Nakkiran, 2024; Chidambaram et al., 2024).

In a compelling study, Kadkhodaie et al. (2023) has demonstrated that as the train dataset size increases diffusion models make a transition from memorizing the train dataset of facial images to a generalization regime where two diffusion models of identical architecture can produce similar looking new faces even when trained on disjoint sampling sets. Yoon et al. (2023) has given a

precise definition of memorization capacity of a diffusion model. However when the model is not memorizing, to what extent it is actually generalizing and sampling from the underlying distribution is an open question that we study in this paper.

Recent advancements toward achieving that objective have been made in (Shah et al., 2023; Cui et al., 2024; Biroli et al., 2024; Wang et al., 2024). Biroli et al. (2024) has studied the evolution of Gaussian mixture model in forward diffusion process from the point of view of the random energy model and analytically established various important time scales in asymptotically large dimensions $d$ with exponentially large number of samples $n = \mathcal{O}(e^d)$. On the other hand, Cui et al. (2024) has studied the reverse diffusion process of finite number of samples $n = \mathcal{O}(d^0)$. In this limit it is shown that the mean of the underlying distribution can be recovered with a trained two-layer denoiser with one hidden neuron and a skip connection up to square error that scales as $d/n$. Similar upper bound on the square error is noted previously by Shah et al. (2023). Very recently Wang et al. (2024) has provided a probabilistic upper bound of order $d/(1 - \sqrt{d/n})^2$ on the square Frobenius error in recovering the variance of the underlying distribution of a diffusion model with strong inductive bias for sufficiently large train dataset $n = \mathcal{O}(d)$. It is not known how tight the upper bound is in the regime of memorization/generalization or when the bound is saturated.

## 2 FOUNDATIONS OF DIFFUSION-DRIVEN GENERATIVE MODELS

### 2.1 REVIEW OF SCORE BASED GENERATIVE PROCESS

In this section, we review basic notions of diffusion based generative models. In particular, we examine an exactly solvable stochastic differential equation (SDE), emphasizing various analytically known data size scales that will be of relevance in subsequent discussions.

The Itô SDE under consideration is known as the Ornstein-Uhlenbeck Langevin dynamics, and is expressed by:
$$dX_t^F = -X_t^F dt + \sqrt{2}dW_t, \quad X_t^F \sim \rho(t). \tag{1}$$

The score function associated with the stochastic process will be denoted as (see Appendix A for details of notation and conventions)

$$s(t, x) = \nabla_x \log \rho(t, x) = \frac{1}{\rho(t, x)} \nabla_x \rho(t, x) \tag{2}$$

The probability density $\rho$ satisfies the transport equation (see (27) in Appendix A)

$$\begin{aligned} \partial_t \rho(t, x) &= \nabla \cdot ((x + s(t, x))\rho(t, x)) \\ &= \nabla^2 \rho(t, x) + x.\nabla \rho(t, x) + d\rho(t, x). \end{aligned} \tag{3}$$

The dimension of the data is defined to be given by $d = \dim(x)$. The time evolution of the probability distribution is exactly solvable and given by

$$\rho(t, X_t^F) = \int dX_0^F \; \rho(0, X_0^F) \, \mathcal{N}(X_t^F | X_0^F e^{-t}, 1 - e^{-2t}). \tag{4}$$

Suppose we know the probability density $\rho(0, x)$ exactly. One way to sample from it would be to use the knowledge of the exact score function $s(t, x)$ in the reverse diffusion process (see (34) in Appendix A), i.e,
$$dX_t^B = (-X_t^B - 2s(t, X_t^B))dt - \sqrt{2}\, dW_{1-t} \tag{5}$$

starting from a late time distribution $\rho(T, x)$ (it is assumed that we know how to sample from $\rho(T, x)$).

In the domain of generative AI, we don't know the exact functional form of $\rho(0, x)$. However we have access to finite number of samples from it and the goal of a diffusion model is to generate more data points from the unknown probability density $\rho(0, x)$. Traditional likelihood maximization technique would assume a trial density function $\rho_\theta$ and try to adjust $\theta$ so that likelihood for obtaining known samples is maximized. In this process determination of the normalization of $\rho_\theta$ is computationally expensive as it requires multi-dimensional integration (typically it is required for each step of the optimization procedure for $\theta$). Score based stochastic method mentioned above is

an alternative (Sohl-Dickstein et al., 2015; Song & Ermon, 2019; Ho et al., 2020). This requires estimating the score function $s$ from known samples - it can be obtained by minimizing Fisher divergence (Hyvärinen, 2005) (see (32) in Appendix A for more details) using techniques such as the kernel method (Sriperumbudur et al., 2017) or denoising score matching (Vincent, 2011).

## 2.2 MIXING TIME-SCALE IS OPTIMAL

In this subsection, we present our first contribution: given $n$ samples from a distribution $\rho(x)$, the mixing time—defined as the minimum diffusion time after which two points exert significant mutual influence—corresponds to the optimal variance of a non-parametric Gaussian density estimator. We now proceed to explain this point in detail.

Our goal is to compare the training probability distribution $\rho_T(x)$[1] inferred based on $n$ samples from $\rho(x) \equiv \rho(0, x)$, original distribution $\rho_O(x)$ inferred from $N \gg n$ samples from the same and the generated distribution $\rho_G$ obtained from a diffusion based generative model (see later sections for more detailed discussion). Before we give precise definition of $\rho_T(x)$ etc. we note certain basic facts about the diffusion process based on finite number of samples. Given $n$ samples $\rho(x)$ can be approximated by the Dirac delta distribution

$$\hat{\rho}(0, x) \equiv \frac{1}{n} \sum_{k=1}^{n} \delta(x - x_k) \tag{6}$$

However the expression of $\hat{\rho}(0, x)$ in terms of Dirac delta function above is singular, rendering traditional metrics of probability divergence, such as Kullback-Leibler divergence, inapplicable. One way to de-singularize it would be to consider $\hat{\rho}(t_M, x)$ where $t_M$ is the lowest time when the contribution from various data points starts getting mixed. The time evolution of the probability distribution under Ornstein-Uhlenbeck diffusion process can be obtained by plugging (6) into (4)

$$\hat{\rho}(t, x) = \frac{1}{n} \sum_{k=1}^{n} \mathcal{N}(x | x_k e^{-t}, 1 - e^{-2t}) \tag{7}$$

To formalize the definition of $t_M$ we consider $x = x_1 e^{-t} + \sqrt{\delta_t} Z$ for some $Z \sim \mathcal{N}(0, \boldsymbol{I}_d)$ and decompose $\rho_T(t, x)$ into parts as follows

$$\hat{\rho}(t, x) = Z_1 + Z_{1^c}, \quad Z_1 = \frac{1}{n} \mathcal{N}(x | x_1 e^{-t}, 1 - e^{-2t}), \quad Z_{1^c} = \frac{1}{n} \sum_{k=2}^{n} \mathcal{N}(x | x_k e^{-t}, 1 - e^{-2t}). \tag{8}$$

It can be shown that in the limit $n \to \infty$ with $\alpha = \log n / d$ fixed, $Z_1 = \mathcal{O}(e^{-d/2}/n)$ and $Z_{1^c}$ is an increasing function of $t$ such that for $t < t_M$, $Z_{1^c} < Z_1$ and at $t = t_M$, $Z_{1^c} = e^{-d/2}/n$. When $\rho(0, x) = \mathcal{N}(x | \mu, \sigma^2 \boldsymbol{I}_d)$, one can explicitly calculate $t_M$ using ideas from random energy model (Gross & Mezard, 1984) to be given by (Biroli et al., 2024)

$$t_M(\sigma, d, n) = \frac{1}{2} \log \left( 1 + \frac{\sigma^2}{n^{\frac{2}{d}} - 1} \right) = \frac{1}{2} \left( \frac{\sigma^2}{n^{\frac{2}{d}}} \right) + \mathcal{O} \left( \frac{\sigma^4}{n^{\frac{4}{d}}} \right) \tag{9}$$

In the second equality, we have further expanded the expression above in the limit of large $\alpha \gg \log(\sigma)$ and kept only the leading order term. With these ideas in mind we define the train distribution to be given by

$$\rho_T(x) \equiv \frac{1}{n} \sum_{k=1}^{n} \mathcal{N}(x | x_k, \epsilon^2) \approx \hat{\rho}(t_M, x), \quad \epsilon^2 = 2 t_M(\sigma, d, n) \ll 1 \tag{10}$$

On a separate line of work, Rosenblatt (1956); Epanechnikov (1969); Bickel & Rosenblatt (1973) has minimized the expectation value of $\int dx (\rho(x) - \rho_T(x))^2$ over $\epsilon$ when $\rho_T$ is given by $\rho_T(x) = (1/n) \sum_{k=1}^{n} \mathcal{N}(x | x_k, \epsilon^2)$ for large $n$ at fixed $d$ to obtain the following formula for the optimal $\epsilon$

$$\epsilon = \left( \frac{4}{d+2} \right)^{\frac{1}{d+4}} \frac{\sigma}{n^{\frac{1}{d+4}}} = \frac{\sigma}{n^{\frac{1}{d}}} + \mathcal{O} \left( \frac{\log d}{d}, \frac{\log n}{d^2} \right) \tag{11}$$

---

[1]Not to be confused with $\rho(T, x) \neq \rho_T(x)$.

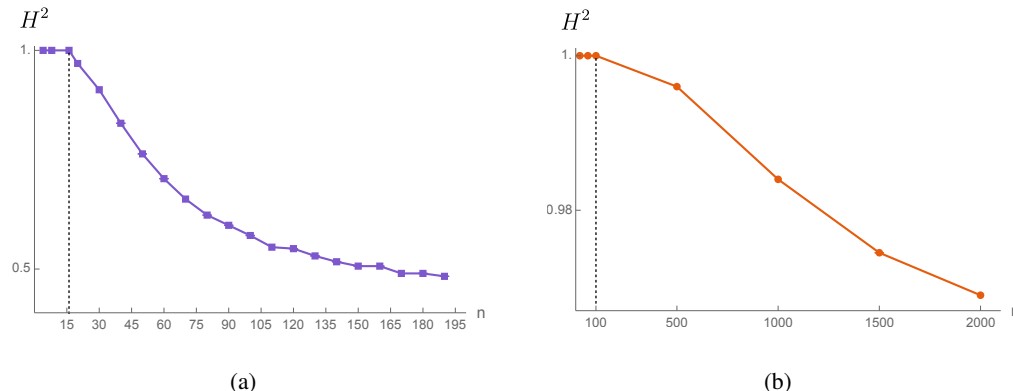

(a)                                                           (b)

Figure 2: The plot is based on the expression of error for the linear diffusion model in (18). The data set is drawn from a Gaussian distribution of mean $\mu = 10$ and diagonal standard deviation $\sigma = 1$ of dimension $d = 16$ (left), 100 (right). We have fixed $T = 2, \lambda = e^{-4}$. We see that the fast convergence begins at around $n \sim d$ as opposed to $n \sim e^d$.

To go to the second equality we have taken $d$ large. Due to the order of limits, this calculation is valid only for large $\alpha$ that does not scale with $n, d$.

We observe that the first term in (11) matches precisely with the expression of $\sqrt{2t_M}$ as calculated above in (9). This shows that the region when both the analytical formula for mixing time of diffusion process $t_M$ and that of the optimal standard deviation of the Gaussian kernel $\epsilon$ is valid, they coincide with each other making the mixing time $L^2$-distance optimal.

To summarize, given a finite number of samples for the underlying distribution we will use it to define the train distribution as in (10) and similar definition applies for the original distribution $\rho_O(x)$.

### 2.3 LATE TIME DISTRIBUTION IS GAUSSIAN

We turn to argue that, at sufficiently late times in the forward diffusion process, it is reasonable to approximate $\hat{\rho}$ given in (7) by a Gaussian distribution. Specifically, at these late times—when $|xe^{-t}| \ll 1$—it becomes convenient to expand the expression of $\hat{\rho}$ as follows

$$\log \hat{\rho}(t, x) = -\frac{d}{2} \log 2\pi \delta_t + \frac{1}{\delta_t} xe^{-t} . \langle y \rangle_{\pi_t} - \frac{1}{2\delta_t} xe^{-t} . \Sigma_t xe^{-t} + \mathcal{O}((xe^{-t})^3), \qquad (12)$$

where the expectations are taken with respect to the density function

$$\pi_t(y) = \rho(0, y) e^{-\frac{y^2 e^{-2t}}{2\delta_t}}, \quad \delta_t = 1 - e^{-2t} \qquad (13)$$

and the quadratic variance matrix is given by

$$(\Sigma_t)_{ij} = e^{2t} \delta_{ij} - (\langle y_i y_j \rangle_{\pi_t} - \langle y_i \rangle_{\pi_t} \langle y_j \rangle_{\pi_t}) \qquad (14)$$

As $t \to \infty$ all the eigenvalues of $\Sigma_t$ are positive. This continues to be the case as long as $t > t_R$, where $t_R$ is a dynamical property of the distribution $\rho$ (Biroli et al., 2024). We will interpret this as follows: for $t > t_R$, we can reliably approximate $\hat{\rho}$ by a suitable Gaussian distribution. In next section we use this idea to propose a linear diffusion model.

### 3 LINEAR DIFFUSION MODEL

In this section we define and study a linear denoiser diffusion model. First, we sample $Y_k, k = 1, 2, .., n$ from the underlying distribution $\rho(x)$ and add noise $Z_k \sim \mathcal{N}(0, \boldsymbol{I}_d)$ to it to obtain noisy samples $X_k = e^{-T} Y_k + \sqrt{\Delta_T} Z_k, \Delta_T = \lambda \delta_T$. Here $T \gg t_R$ is a large enough time scale and $\lambda$

is a free hyperparameter that controls the amount of noise added [2]. For simplicity we add the entire noise in one step in contrast to multi-step process of realistic diffusion models.

The diffusion model/denoiser, trained on the data above, as input takes a noisy sample $X$ and generates a clean sample $Y$. In this paper, we consider a linear model

$$Y = \hat{\theta}_0 + \hat{\theta}_1 X$$

as prototype denoiser for analytical tractability. The parameters $\hat{\theta}_0, \hat{\theta}_1$ are solution to the standard linear regression problem of predicting $\{Y_k\}$ given $\{X_k\}$ and given by

$$\hat{\theta}_1^T = (x^T x)^{-1} x^T y, \quad \hat{\theta}_0 = \hat{Y} - \theta_1 \hat{X} \quad \hat{X} = \frac{1}{n} \sum_{k=1}^n X_k, \quad \hat{Y} = \frac{1}{n} \sum_{k=1}^n Y_k \tag{15}$$

Here $x, y$ are $n \times d$ dimensional matrices whose $k$-th row is $(X_k - \hat{X})^T, (Y_k - \hat{Y})^T$ respectively.

To generate samples from the trained diffusion model we first draw $X$ from $\mathcal{N}(\mu_X, \sigma_X^2 \boldsymbol{I}_d)$, motivated by the fact that late time distribution can be reliably approximated by a Gaussian distribution, with

$$\mu_X = e^{-T}\hat{Y}, \quad \sigma_X^2 = e^{-2T}\frac{1}{nd}\sum_{k=1}^n ||Y_k - \hat{Y}||^2 + \Delta_T, \quad \hat{Y} = \frac{1}{n}\sum_{k=1}^n Y_k \tag{16}$$

and then use the diffusion model to predict corresponding $Y = \hat{\theta}_0 + \hat{\theta}_1 X$. The generated probability distribution for a given set $\{(X_k, Y_k), k = 1, 2, .., n\}$ is

$$\rho_G(Y|\{(X_k, Y_k), k = 1, 2, .., n\}) = \mathcal{N}(Y|\hat{\theta}_0 + \hat{\theta}_1 \mu_X, \sigma_X^2 \hat{\theta}_1^T \hat{\theta}_1) \tag{17}$$

The generated probability distribution $\rho_G$ as defined above is a random variable conditioned on $\{(X_k, Y_k), k = 1, 2, .., n\}$ which itself is a random variable. We consider its expectation value by further sampling $Y_k \sim \rho$ and $X_k = e^{-T}Y_k + \sqrt{\Delta_T}Z_k$ as mentioned above. The sampling procedure mentioned here is different from the discussion of standard linear regression. In fact the unconditioned distribution $\rho_G(Y)$ obtained after taking into account the sampling of $\{(X_k, Y_k), k = 1, 2, .., n\}$ is a complicated one.

## 3.1 CONVERGENCE OF THE GENERATED PROBABILITY DISTRIBUTION

The convergence of the generated distribution to the original distribution is measured by the Hellinger distance

$$\begin{aligned}
& \mathrm{H}^2(\rho||\rho_G|\{(X_k, Y_k), k = 1, 2, .., n\}) \\
=& \frac{1}{2}\int dx\, (\sqrt{\rho(x)} - \sqrt{\rho_G(x)})^2 = \int dx\, (1 - \sqrt{\rho(x)\rho_G(x)}) \\
=& 1 - \Big(\frac{2^d\sqrt{\det(2\pi\sigma_X^2\hat{\theta}_1^T\hat{\theta}_1)\det(2\pi\sigma^2\boldsymbol{I}_d)}}{\det\Big(2\pi(\sigma_X^2\hat{\theta}_1^T\hat{\theta}_1 + \sigma^2\boldsymbol{I}_d)\Big)}\Big)^{\frac{1}{2}}e^{-\frac{1}{4}(\hat{\theta}_0 + \hat{\theta}_1\mu_X - \mu)^T(\hat{\theta}_1^T\hat{\theta}_1 + \sigma^2\boldsymbol{I}_d)^{-1}(\hat{\theta}_0 + \hat{\theta}_1\mu_X - \mu)}
\end{aligned} \tag{18}$$

The smaller the value of $H^2$ the closer the generated distribution is to the original one.

For simplicity we sample $Y_k$ from an isotropic Gaussian distribution $Y_k \sim \mathcal{N}(\mu, \sigma^2\boldsymbol{I}_d)$. We have plotted the expectation value of $H^2$ in Fig. 2. We notice that with the increase of the train data size, there is a domain of fast convergence marked by a sharp decay of $H^2$ with higher slope beyond a critical value. The occurrence of fast convergent region is conceptually similar to the sharp improvement of the performance of a trained denoiser on a test dataset as the size of the training set increases above a certain threshold emprically observed in (Kadkhodaie et al., 2023).

---

[2]This corresponds to scaling the noise term in (1) by a factor of $\sqrt{\lambda}$.

### 3.2 MEMORIZATION TO GENERALIZATION TRANSITION

In this subsection we define and analyze a rigorous metric for memorization to generalization transition. For this purpose consider the pairwise distance between probability distributions $\rho_O, \rho_T, \rho_G$

$$
\begin{aligned}
\mathrm{E}_{TG} &= \int dx \, (\rho_T(x) - \rho_G(x))^2, \\
\mathrm{E}_{OG} &= \int dx \, (\rho_O(x) - \rho_G(x))^2.
\end{aligned}
\tag{19}
$$

We say the model is generalizing well if the generated data distribution is closer to the original dataset compared to the train data, more concretely it is defined by the following condition

$$
\Delta = \mathrm{E}_{TG} - \mathrm{E}_{OG} > 0.
\tag{20}
$$

$\Delta$ is a random variable due to randomness in the training data. The diffusion model is considered to be *memorizing* the training data when the probability that $\Delta > 0$ satisfies $P(\Delta > 0) \le 0.5$. Conversely, the model is in the *generalization regime* when $P(\Delta > 0) > 0.5$. For the linear diffusion model, the pairwise distance between probability distributions $\rho_O, \rho_T, \rho_G$ are easily calculated to be given by

$$
\begin{aligned}
E_{TG}(\{(X_k, Y_k), k = 1, 2, .., n\}) &= \frac{1}{\sqrt{\det(4\pi\hat{\theta}_1^T \hat{\theta}_1 \sigma_X^2)}} \\
&\quad - 2\sum_{k=1}^{n} \frac{e^{-\frac{1}{2}(\hat{\theta}_0 + \hat{\theta}_1 \mu_X - Y_k)^T (\hat{\theta}_1^T \hat{\theta}_1 \sigma_X^2 + \epsilon^2 \boldsymbol{I}_d)^{-1} (\hat{\theta}_0 + \hat{\theta}_1 \mu_X - Y_k)}}{n\sqrt{\det\left(2\pi(\hat{\theta}_1^T \hat{\theta}_1 \sigma_X^2 + \epsilon^2 \boldsymbol{I}_d)\right)}} \\
&\quad + \frac{n + 2\sum_{i,j=1, i<j}^{n} e^{-\frac{1}{2}(Y_i - Y_j)^T (2\epsilon^2 \boldsymbol{I}_d)^{-1}(Y_i - Y_j)}}{n^2\sqrt{\det(4\pi\epsilon^2 \boldsymbol{I}_d)}} \\
E_{OG}(\{(X_k, Y_k), k = 1, 2, .., n\}) &= \frac{1}{\sqrt{\det(4\pi\hat{\theta}_1^T \hat{\theta}_1 \sigma_X^2)}} \\
&\quad - 2\sum_{k=1}^{N} \frac{e^{-\frac{1}{2}(\hat{\theta}_0 + \hat{\theta}_1 \mu_X - Y_k)^T (\hat{\theta}_1^T \hat{\theta}_1 \sigma_X^2 + \epsilon^2 \boldsymbol{I}_d)^{-1} (\hat{\theta}_0 + \hat{\theta}_1 \mu_X - Y_k)}}{N\sqrt{\det\left(2\pi(\hat{\theta}_1^T \hat{\theta}_1 \sigma_X^2 + \epsilon^2 \boldsymbol{I}_d)\right)}} \\
&\quad + \frac{N + 2\sum_{i,j=1, i<j}^{N} e^{-\frac{1}{2}(Y_i - Y_j)^T (2\epsilon^2 \boldsymbol{I}_d)^{-1}(Y_i - Y_j)}}{N^2\sqrt{\det(4\pi\epsilon^2 \boldsymbol{I}_d)}}
\end{aligned}
\tag{21}
$$

For given $Y_k, k = 1, 2, \ldots, n$ we calculate $X_k$ as

$$
Y_k \sim \mathcal{N}(\mu, \sigma^2 \boldsymbol{I}_d), \quad X_k = e^{-T} Y_k + \sqrt{\Delta_T} Z_k \quad Z_k \sim \mathcal{N}(0, \boldsymbol{I}_d), \quad T \gg t_R
\tag{22}
$$

By performing several simulations over the training set $\{(X_k, Y_k), k = 1, 2, .., n\}$ we have plotted the average value of the probability of $\Delta > 0$ in Fig. 3. The important regions for $n$ are the following:

- $N \gg d \gg n$: As we increase $n$, probability of $\Delta > 0$ increases sharply from zero to one. We call this memorization to generalization transition. See Fig. 3 for more details.

- $N \gg n \sim d$: In this domain $P(\Delta > 0)$ saturates near unity at $n \sim d$ as shown in Fig. 3. From Fig. 2 we see that near $n \sim d$, we enter at the regime of fast convergence where the Hellinger distance between the original and generated distribution $H^2$ decreases rapidly. This shows the initiation of generalization (i.e., $P(\Delta > 0) \sim 1$) takes place on the onset of fast convergence.

- $N \sim n \gg d$: In this domain $P(\Delta > 0)$ decreases and eventually reaches the value 0.5. See Fig. 3 for further details. This is because the distinction between the train and the original dataset disappears in this limit.

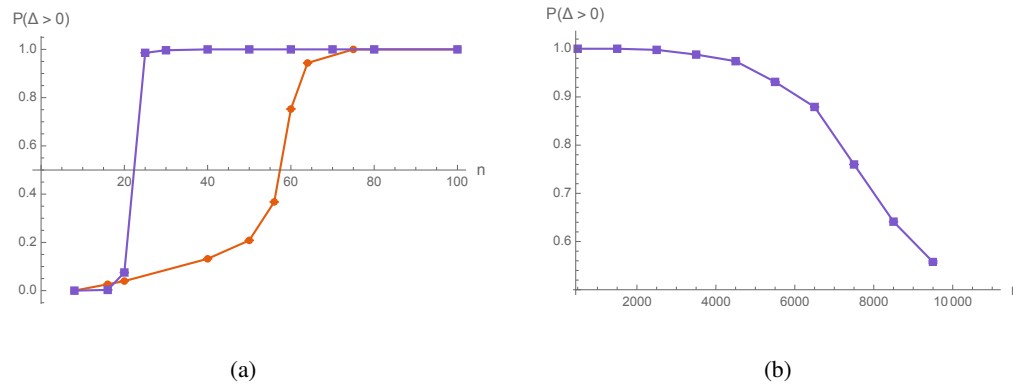

(a)                                                                 (b)

Figure 3: The plot on the left is based on the expressions for the linear diffusion model in (21). The data distribution and the model configuration is the same as in Fig. 2. The original dataset is composed of $N = 10^4$ samples. We de-singularized the train and the original distribution with $\epsilon = \sigma(1/n)^{1/d}, \sigma(1/N)^{1/d}$ respectively - see the formula in (9). Comparing with Fig. 2 we notice that the model learns to generalize when the convergence of the generated probability distribution towards the original distribution becomes rapid. On right we have plotted the curve for much larger train dataset size for $d = 16$. It is observed that as the size of the train dataset $n$ approaches that of the original dataset $N = 10^4$, $P(\Delta > 0)$ converges towards 0.5, indicating that the distinction between the training and original datasets becomes less prominent.

## 4  NEURAL NETWORK BASED DIFFUSION MODEL

Following the results of the linear diffusion model, we hypothesize that memorization to generalization transition happens in generic diffusion model on the onset of fast convergence. In this section we test our hypothesis on more realistic models. We use the PyTorch based implementation of the algorithm in Ho et al. (2020) as the diffusion model for the experiments in this section. The denoiser has the structure of U-Net (Ronneberger et al., 2015) with additional residual connections consisting of positional encoding of the image and attention layers (Dosovitskiy et al., 2021; Tu et al., 2022; Peebles & Xie, 2023). Our code is attached as supplementary files with this draft. The generated distribution from the non-linear diffusion model is not necessarily Gaussian and we face difficulty in evaluating the higher dimensional integration in our measure of distance. For this purpose we use the following simplified metric which gives advantage in terms of computational complexity

$$\mathrm{E}_{TG} = \sum_{i=1}^{d} \int dx \, (\rho_{T,i}(x) - \rho_{G,i}(x))^2 \tag{23}$$

Where we have defined element-wise probability density function

$$\rho_{T,i}(x) \equiv \frac{1}{n} \sum_{k=1}^{n} \mathcal{N}(x|x_k^i, \epsilon^2), \quad \epsilon = \frac{\sigma}{n}, \quad i = 1, 2, \dots, d \tag{24}$$

Similar definition applies to $\rho_{O,i}, \rho_{G,i}$ and $E_{OG}$. The convergence and generalization metric for an isotropic Gaussian dataset is plotted in Fig. 4. We see that memorization to generalization transition takes places exactly on the onset of rapid convergence of the generalization error $E_{OG}$.

## 5  CONCLUSION

In this paper we have defined a mathematically precise metric for memorization to generalization transition. We have further constructed a linear diffusion model which shows the memorization to generalization transition, based on the aforementioned metric, aligns with the on-set of fast convergence of generalization error. Finally we have shown that this alignment phenomena is generic and occurs in realistic non-linear diffusion models.

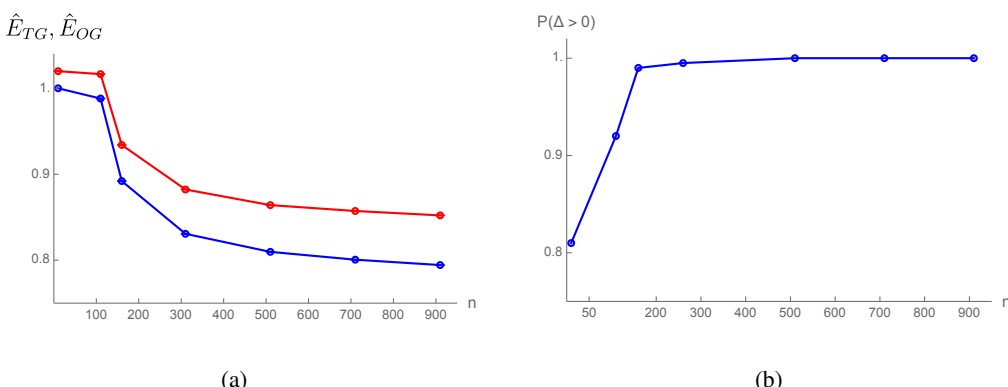

(a)                                                                                   (b)

Figure 4: The plot is based on the neural network based diffusion model with 12.9 million trainable parameters. Each pixel is a Gaussian with centers at $\mu = 0.5$ and standard deviation $\sigma = 0.1$ of dimension $d = 256$. The number of samples in the original dataset is $10^5$. The train probability density is calculated with $\epsilon = \sigma/n$, similarly for generated and original distribution we use appropriate scale. To calculate $\Delta$ we have generated 10 images and compared against train and original dataset of size $10, 100$ (randomly selected) in each simulation. The numerical integral is performed by summing over $|S| = 2^4$ equidistant points in $(0, 1)$ for each pixel. Training is done for 20 epoch with batch size 128 and diffusion step number is kept fixed at 10. $\hat{E}_{OG}$ is the scaled value of $E_{OG}$ by its value at $n = 10$ measuring the convergence of the generated distribution to the original one. We see that the value of $\hat{E}_{OG}$ decreases rapidly (on left in blue) during sharp increase of $P(\Delta > 0)$ marking memorization to generalization transition (on right). The scaled value of $\hat{E}_{TG}$ is also plotted in red on left.

Our work generates the possibility of further analytical study on diffusion models. For instance, instead of a linear diffusion model, a (stack of) wide neural network in the kernel approximation regime (Jacot et al., 2018; Lee et al., 2019; Bordelon et al., 2021; Canatar et al., 2021; Atanasov et al., 2023) or in mean field regime (Mei et al., 2018; Yang & Hu, 2021; Bordelon & Pehlevan, 2022; 2024) is an alternate viable candidate. Furthermore, the proposed metric for memorization to generalization transition might be employed to systematically select optimal hyperparameters for model training. Specifically, for a given training set, one can plot the curve of $P(\Delta > 0)$ versus $n$ across various hyperparameter configurations. The model that transitions to generalization at a smaller dataset size may be regarded as the more efficient one. We leave a thorough investigation of these questions to future work.

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

## A CONNECTION BETWEEN ODE AND SDE BASED GENERATIVE MODELS

In this Appendix we review the connection between stochastic interpolant (Albergo et al., 2023) and stochastic differential equation (Song et al., 2021b) based generative models. Given two probability density functions $\rho_0, \rho_1$, one can construct a stochastic interpolant between $\rho_0$ and $\rho_1$ as follows

$$x(t) = X(t, x_0, x_1) + \lambda_0(t)z, \qquad t \in [0, 1] \tag{25}$$

where the function $X, \lambda_0$ satisfies

$$X(0, x_0, x_1) = x_0, \quad X(1, x_0, x_1) = x_1, \quad ||\partial_t X(t, x_0, x_1)|| \leq C||x_0 - x_1|| \\ \lambda_0(0) = 0, \quad \lambda_0(1) = 0, \quad \lambda_0(t) \geq 0 \tag{26}$$

for some positive constant $C$. Here $x_0, x_1, z$ are drawn independently from a probability measure $\rho_0$, $\rho_1$ and standard normal distribution $\mathcal{N}(0, \boldsymbol{I})$. The probability distribution $\rho(t, x)$ of the process $x(t)$ satisfies the transport equation[3]

$$\partial_t \rho + \nabla \cdot (b\rho) = 0, \quad \rho(0, x) = \rho_0(x), \quad \rho(1, x) = \rho_1(x), \tag{27}$$

where we defined the velocity[4]

$$b(t, x) = \mathbb{E}[\dot{x}(t)|x(t) = x] = \mathbb{E}[\partial_t X(t, x_0, x_1) + \dot{\lambda}_0(t)z|x(t) = x]. \tag{28}$$

One can estimate the velocity field by minimizing

$$L_b[\hat{b}] = \int_0^1 \mathbb{E}\left(\frac{1}{2}||\hat{b}(t, x(t))||^2 - \left(\partial_t X(t, x_0, x_1) + \dot{\lambda}_0(t)z\right) \cdot \hat{b}(t, x(t))\right) dt \tag{29}$$

It's useful to introduce the score function $s(t, x)$ for the probability distribution for making the connection to the stochastic differential equation

$$s(t, x) = \nabla \log \rho(t, x) = -\lambda_0^{-1}(t)\mathbb{E}(z|x(t) = x) \tag{30}$$

It can be estimated by minimizing

$$L_s[\hat{s}] = \int_0^1 \mathbb{E}\left(\frac{1}{2}||\hat{s}(t, x(t))||^2 + \lambda_0^{-1}(t)z \cdot \hat{s}(t, x(t))\right) dt \tag{31}$$

The score function also can be obtained by minimizing the following alternative objective function known as the Fisher divergence

$$L_F[\hat{s}] = \frac{1}{2}\int_0^1 \mathbb{E}\left(||\hat{s}(t, x(t)) - \nabla \log \rho(t, x)||^2\right) dt$$
$$= \int_0^1 \mathbb{E}\left(\frac{1}{2}||\hat{s}(t, x(t))||^2 + \nabla \cdot \hat{s}(t, x(t)) + \frac{1}{2}||\nabla \log \rho(t, x))||^2\right) dt \tag{32}$$

To obtain the second line we have ignored the boundary term. Note that for the purpose of minimization the last term is a constant and hence it plays no role and hence Fisher divergence can be minimized from a set of samples drawn from $\rho$ easily even if the explicit form of $\rho$ is not known (Hyvärinen, 2005). However the estimation of $\nabla \cdot \hat{s}(t, x(t))$ is computationally expensive and in practice one uses denoising score matching for estimating the score function (Vincent, 2011).

It is easy to put eq. (27) into Fokker-Planck-Kolmogorov form

$$\partial_t \rho + \nabla \cdot (b_F \rho) = +\lambda(t)\Delta\rho, \qquad b_F(t, x) = b(t, x) + \lambda(t)s(t, x)$$
$$\partial_t \rho + \nabla \cdot (b_B \rho) = -\lambda(t)\Delta\rho, \qquad b_B(t, x) = b(t, x) - \lambda(t)s(t, x) \tag{33}$$

For an arbitrary function $\lambda(t) \geq 0$. From this we can read off the Itô SDE as follows[5]

$$dX_t^F = b_F(t, X_t^F)dt + \sqrt{2\lambda(t)}\, dW_t$$
$$dX_t^B = b_B(t, X_t^B)dt - \sqrt{2\lambda(t)}\, dW_{1-t} \tag{34}$$

First equation is solved forward in time from the initial data $X_{t=0}^F \sim \rho_0$ and the second one is solved backward in time from the final data $X_{t=1}^B \sim \rho_1$. One can recover the probability distribution $\rho$ from the SDE using Feynman–Kac formulae[6]

$$\rho(t, x) = \mathbb{E}\left(e^{\int_t^0 \nabla \cdot b_F(t, Y_t^B)dt}\rho_0(Y_{t=0}^B)|Y_t^B = x\right)$$
$$= \mathbb{E}\left(e^{\int_t^1 \nabla \cdot b_B(t, Y_t^F)dt}\rho_1(Y_{t=1}^F)|Y_t^F = x\right) \tag{36}$$

---

[3]Here we are using the notation $\nabla = \nabla_x$.

[4]The expectation is taken independently over $x_0 \sim \rho_0, x_1 \sim \rho_1$ and $z \sim \mathcal{N}(0, \boldsymbol{I})$. Here $\mathcal{N}(0, \boldsymbol{I})$ is normalized Gaussian distribution of appropriate dimension with vanishing mean and variance.

[5]Here $W_t$ represents a standard Wiener process, i.e., $W_t - tW_1 = N_t$ is a zero-mean Gaussian stochastic process that satisfies $\mathbb{E}[N_t N_t^\top] = t(1-t)\boldsymbol{I}$.

[6]A class of exactly solvable models are given by (Ornstein-Uhlenbeck dynamics discussed in the main text is a special case of this equation)

$$dX_t^F = X_t^F \frac{d}{dt}(\log \eta(t))dt + \sqrt{\eta(t)^2 \frac{d}{dt}\left(\frac{\sigma(t)^2}{\eta(t)^2}\right)}dW_t, \quad X_t^F \sim \mathcal{N}(\eta(t)X_0^F, \sigma(t)^2) \tag{35}$$

Where $\eta, \sigma$ are two positive functions satisfying $\eta(0) = 1, \sigma(0) = 0$.

