# OpenReview forum: "On the onset of memorization to generalization transition in diffusion  models"
_ICLR.cc/2025/Conference — ICLR 2025 Conference Withdrawn Submission_

### Official Review · Reviewer_fE6V · 2024-10-30

**Soundness:** 2
**Presentation:** 1
**Contribution:** 1
**Rating:** 3
**Confidence:** 4

**Summary:**

This paper studies the memorization-to-generalization transition of a linear diffusion model in the case where the underlying distribution is Gaussian. They provide quantitative analysis of sample complexity (train set size) threshold for which the model-generated distribution from the linear diffusion model for Gaussian distribution starts to become closer to the underlying distribution than the train set distribution.

**Strengths:**

The authors provide a solid statistics-based discussion on the Gaussian data & linear diffusion model scenario.

**Weaknesses:**

My main concern is that the paper lacks a key result or message that (at least potentially) has meaningful implications on large-scale diffusion model training. Another problem is that the paper's presentation is sloppy in many aspects and makes the paper unnecessarily difficult to read through. To me, this paper seems to require a lot of development and re-organization to be ready for publication.

1. One of the main contributions that the authors claim is to identify the pattern of transition from memorization to generalization. However, while this heavily relies on the properties of data distribution, their theoretical analysis only considers the Gaussian distribution. It is entirely unclear whether the same pattern would be observed from real datasets. For example, in the abstract the authors say "This is to be contrasted with an alternative scenario, where the model's memorization performance degrades, but generalization performance doesn't improve.", but I think such regime can be found with real datasets (see, e.g., [1]).

2. A similar, additional weakness emerges from assuming the linear diffusion model, where the sampling procedure does not take into account the numerical discretization procedure of SDE/ODE, which is the central component of score-based generative modeling. They do provide a simple experiment using more realistic architecture and sampling procedure, but it does not necessarily indicate that the paper's finding applies to the nonlinear sampling procedure because the underlying data distribution is isotropic Gaussian (in this case, all levels of noised distributions are Gaussian and the corresponding score functions linear. Therefore, linear and nonlinear sampling procedures do not seem fundamentally different, if not identical).

3. The observation that rapid decrease in $E_{OG}$ is aligned with generalization seems like a direct implication of definition, and is not surprising. Why is this an interesting finding?

[1] Feng et al., When do GANs replicate? On the choice of dataset size. ICCV, 2021.

**Questions:**

- In page 3, line 150, the authors write "Suppose we know ... $\rho(0,x)$ exactly." and then describe the score-based sampling procedure. However, the presentation is misleading because knowing $\rho(0,x)$ exactly does not necessarily enable the score-based (diffusion model) sampling. It rather requires the knowledge of $\rho(T,x)$ and the score functions $s(t,x)$.
- In page 4, line 174, the authors write "original distribution $\rho_O(x)$ inferred from $N \gg n$ samples". What does this mean? While they seem to use the notation $N$ for the first time here, it is not even properly defined, and it is unclear what $\rho_O$ precisely is and why it should be distinguished with $\rho$.
- The authors do not explicitly state that the samples $x_k$ belong to $\mathbb{R}^d$, which will confuse the readers.
- In page 4, line 196, I do not understand what the limit $n \to \infty$ with $\alpha = \log n / d$ fixed means. Here they are considering the limits of $Z_1$ and $Z_{1^c}$, which rely on $x_1 \in \mathbb{R}^d$. Then, does it make sense to take the limit while $d$ is changing?
- In line 197, what does $Z_{1^c} = e^{-d/2}/n$ mean? Isn't the left hand side dependent on $x$? Is $\mathcal{O}$ missing?
- What is the precise meaning of $\approx$ symbol in equation (10)?
- What does the notation $\mathcal{O}(\cdot, \cdot)$ in equation (11) mean?
- In page 5, equation (14), $y_i, y_j$ appears suddenly. Are they the train set samples? If so, they should have been specified earlier. It is very confusing because in the previous page, the train samples were denoted as $x_k$.
- In page 6, the meaning of lines 298-301 is unclear.
- In page 7, line 337-338, the definition "The diffusion model is considered to be memorizing the training data when the probability that $\Delta > 0$ ..." is confusing. The diffusion model already relies on the train data, but the probability is with respect to the randomness in train data. It does not make sense to discuss the memorization of an individual diffusion model in this way. I think this should rather be the property of the _dataset_, not the _model_.

---

> ### Author Response · Authors · 2024-11-25
>
> We thank the reviewer for their feedback and will respond to the raised questions below.
>
> **1. Reply to "My main concern is that the paper lacks a key result or message that (at least potentially) has meaningful implications on large-scale diffusion model training."**
>
> If diffusion model A makes a transition from memorization to generalization and sample from the underlying distribution at a smaller train dataset size compared to diffusion model B, then clearly model A is preferable. In this paper, we provide a clear metric for the transition. We hope these developments will eventually have meaningful effects on how to train diffusion models optimally. We mention this on line 463 of our paper.
>
> **2. Reply to "The observation that rapid decrease in is aligned with generalization seems like a direct implication of definition, and is not surprising. Why is this an interesting finding?"**
>
> According to our definition, memorization to generalization transition takes place when $P(\Delta=E_{TG}-E_{OG}>0)$ changes sharply from a smaller value to close to being one. Even if this is the case, it is one condition on two variables $E_{TG}, E_{OG}$, hence mathematically there are several possibilities for individual behavior of $E_{TG}$ and $E_{OG}$ on the onset of the transition.
> We have explicitly drawn some of the possibilities in figure 1. One possibility is that $E_{TG}$ increases while $E_{OG}$ decreases as the size of the training dataset is increased featuring a sharp transition in $P(\Delta>0)$. For the realistic diffusion model on the Gaussian dataset, we find that this is not the case.
>
> **Answer to Q1:**
>
> We thank the reviewer for this comment.
>
> **Answer to Q2:**
>
> It means taking $N$ samples from the underlying true distribution $\rho$ and then using our Kernel prescription as given in equation (10) to construct a probability distribution. We call this probability distribution the original distribution $\rho_O$. In practice, we always have a finite number of samples, which motivates this definition. The distribution $\rho_O$ approaches $\rho$ when $N \to \infty$.
>
> **Answer to Q3:**
>
> We thank the reviewer for this comment.
>
> **Answer to Q4:**
>
> It means that $n=\mathcal{O}(e^{\alpha d}), d\to \infty$. Taking limits of $n,d$ both large keeping some suitable function of them fixed is usual in the study of higher dimensional regression problems for example .
>
> **Answer to Q5:**
>
> Yes.
>
> **Answer to Q6:**
>
> It means we are ignoring higher order terms in $\epsilon$ which is a small number.
>
> **Answer to Q7:**
>
> $\mathcal{O}(a,b)$ means the leading correction is either $\mathcal{O}(a)$ or order $\mathcal{O}(b)$.
>
> **Answer to Q8:**
>
> In equation (14) $y_i$ is just a dummy variable to be integrated over with density $\pi_t$ defined in equation (13).
>
> **Answer to Q9:**
>
> We thank the reviewer for this comment.
>
> **Answer to Q10:**
>
> We agree that the trained diffusion model depends on the dataset and hence its memorization ability depends on it as well.

---

### Official Review · Reviewer_YWqz · 2024-11-04

**Soundness:** 3
**Presentation:** 3
**Contribution:** 2
**Rating:** 5
**Confidence:** 3

**Summary:**

This paper examines the transition from memorization to generalization in diffusion models as the training set size increases. The authors introduce a novel mathematical framework to define and study this transition, identifying when diffusion models stop merely replicating training data and begin to produce novel outputs that resemble the underlying data distribution. Key contributions include a new metric for measuring the transition and empirical evidence that supports the theory in realistic diffusion models.

**Strengths:**

The paper's originality is evident in its novel approach to defining and measuring the memorization-to-generalization transition. The writing is clear. This work has the potential to influence future research directions and practices in training diffusion models, especially regarding the phase transition from memorization to generalization.

**Weaknesses:**

1. The assumption of isotropic Gaussian distribution seems too strong. The gap between the real-world data and this assumption is notable.
2. More experimental details could be provided to explain the empirical results (more elaborated figures etc.).

**Questions:**

**1.** The figures could be better. For example, labels could be added to Figure 3 to clearly describe what the blue and red curve means respectively.

**2.** Figure 4 looks confusing to me. From the left figure, it seems that E_{TG} and E_{OG} are both decreasing at a similar rate, without crossing each other, which is quite different from the illustrated figure in Figure 1(c). How does it make the transition from memorization to generalization?

**3.** Could the assumption of isotropic Gaussian be loosened to non-isotropic Gaussian, or be generalized to Gaussian mixture? It seems that the current assumption is too strong that it has a notable gap from practice.

**4.** Is there a way people can quantify the time of transition in practice based on your analysis? For example, can these metrics of error be well-estimated asymptotically or non-asymptotically?

---

> ### Author Response · Authors · 2024-11-25
>
> We thank the reviewer for their feedback and will respond to the raised questions below.
>
> **1. Reply to "For example, labels could be added to Figure 3 to clearly describe what the blue and red curve means respectively."**
>
> We thank the reviewer for carefully reading our paper and insightful comments. The blue, red curve in figure 3(a) corresponds to data dimension $d=16,100$ respectively.
>
> **2. Reply to "Figure 4 looks confusing to me."**
>
> In figure 4(a), the value of  $\Delta$ at $n=100$ is smaller compared to its value at $n=300$. This fact is marked by the sharp increase in $P(\Delta>0)$ as $n$ is increased from $100$ to $300$. We would also like to emphasize that the metric used for figure (4) is given in equations (23), (24). This metric is similar to the one in equations (10), (19) that is used for the linear diffusion model however, they are not exactly the same. The simpler metric in equations (23), (24) is used to make the computation faster in the realistic diffusion model. As a result of this change in metric, a slightly different numerical value for $P(\Delta>0)$ is visible in figure (4).
>
> **3. Reply to "Could the assumption of isotropic Gaussian be loosened to non-isotropic Gaussian, or be generalized to Gaussian mixture?"**
>
> We thank the reviewer for mentioning this question. We will address it in the future.
>
> **4. Reply to "For example, can these metrics of error be well-estimated asymptotically or non-asymptotically?"**
>
> This is a beautiful question. We most certainly hope that some theoretical results for the linear diffusion model can be established in large $n,d$ limit.

---

### Official Review · Reviewer_r8Ns · 2024-11-05

**Soundness:** 3
**Presentation:** 2
**Contribution:** 2
**Rating:** 5
**Confidence:** 3

**Summary:**

The paper investigates the transition from memorization to generalization in diffusion models as the training set size increases. It defines this transition mathematically with a Gaussian kernel approximation to the training dataset. The study demonstrates a sharp transition in generalization performance when the training set size equals the input dimension for the data from an isotropic Gaussian distribution. Empirical evidence indicates that this phenomenon holds for real diffusion models, where training scale influences memorization and generalization behavior.

**Strengths:**

The paper provides a rigorous analysis to distinguish memorization and generalization regimes in diffusion models, particularly under the assumption of Gaussian-distributed training data.
It observes a distinct transition point when the number of training samples equals the input dimension, which is a compelling theoretical finding that may inspire further study in model behavior with scaling datasets.

**Weaknesses:**

1. The analysis is based on simplified assumptions (Gaussian-distributed data and the SDE process), and the finding may relate more to concepts around mixing time rather than specific diffusion model behavior, a bit hard to translate to more complex, real-world data or standard loss functions like score-function matching.

2. The conclusion may be misleading, for example, the transition point is only related with the input dimension. However, people observe memorization when the dataset size is several order larger than the input dimension (the input image dimension would be ~1e5,  the dataset size 1e7 for imagenet). Moreover, how is the transition related to the data complexity, like facial data is much simpler than the natural images?

3. Notation not explained: $x.\nabla$, $\langle y \rangle_{\pi_t}$, $\langle y_i y_j \rangle_{\pi_t}$, etc.

**Questions:**

There is ambiguity about whether the transition observed is dependent on the input dimension or the intrinsic dimensionality of the data, which may impact the generality of the conclusions.

---

> ### Author Response · Authors · 2024-11-25
>
> We thank the reviewer for their feedback and will respond to the raised questions below.
>
> **Reply to "There is ambiguity about whether the transition observed is dependent on the input dimension or the intrinsic dimensionality of the data, which may impact the generality of the conclusions."**
>
> It is a very important question. However, since our analysis is limited to isotropic Gaussian data, it is not clear to us how the transition point would change for more complicated images.
>
> **Reply to "Notation not explained:.."**
>
> We thank the reviewer for carefully reading our paper and valuable comments.

---

### Official Review · Reviewer_YPdH · 2024-11-06

**Soundness:** 4
**Presentation:** 2
**Contribution:** 1
**Rating:** 3
**Confidence:** 3

**Summary:**

This work analyzes into how diffusion models transition from replicating training data to generalizing from the learned data distributions. This transition is mathematically characterized as the point where the ETG (Error Training to Generated) is less than the EOG (Error Original to Generated). This is a sensible a new novel view of generalization, and the paper provides some analysis based on linear diffusion models.

**Strengths:**

This work considers an important problem of analyzing the conditions that allow a diffusion model to generalize beyond memorizing the training data.

**Weaknesses:**

The analysis of this work is limited to linear diffusion models, and there is almost no experimental validation justifying the relevance of the linear toy model with respect to the practical deep learning setups. The theoretical analysis also does not feel sufficiently surprising and novel enough to warrant a publication at ICLR. There is not much in the paper, so there is perhaps not much to criticize. In my view, the magnitude of the contribution is clearly below the publication expectation of ICLR.

**Questions:**

.

---

> ### Author Response · Authors · 2024-11-25
>
> We thank the reviewer for their feedback and will respond to the raised questions below.
>
> **1. Reply to "This transition is mathematically characterized as the point where the ETG (Error Training to Generated) is less than the EOG (Error Original to Generated)."**
>
> We thank the reviewer for carefully reading our paper. We would like to kindly note that the diffusion model is said to generalize well when the probability of ETG>EOG is close to one.
>
> **2. Reply to "The analysis of this work is limited to linear diffusion models, and there is almost no experimental validation justifying the relevance of the linear toy model with respect to the practical deep learning setups."**
>
> In section 4 we provide experimental evidence in 12.9 million parameter non-linear realistic model justifying our claims. Please see figure 4 for the results in this case.
>
> **3. Reply to "The theoretical analysis also does not feel sufficiently surprising and novel enough to warrant a publication at ICLR."**
>
> One can try to approximate theoretical aspects of more realistic diffusion models in terms of a collection of denoisers, each of which performs a Kernel regression. In this setup, each denoiser performs a linear map in the feature space. The linear diffusion model, in current work, begins theoretical work on just one linear denoiser as a stepping stone for further explorations in the future. Please note that, as of now, we even lack a proper theoretical understanding of this single linear diffusion model to explain the origin of memorization to generalization transition. Our work attempts to fill this gap by providing a solid mathematical framework for the Gaussian probability distribution.

---

### Official Review · Reviewer_s7w1 · 2024-11-09

**Soundness:** 1
**Presentation:** 1
**Contribution:** 1
**Rating:** 1
**Confidence:** 4

**Summary:**

The paper examines the transition from memorization to generalization in diffusion models as the number of training points increases. The authors model the training distribution as a Gaussian-smoothed version of the empirical distribution and define two regimes: memorization and generalization. The model is considered to be in the memorization regime if the generated distribution is closer to the smoothed empirical distribution and in the generalization regime if it is closer to the true population distribution. The paper uses Gaussian data and a linear diffusion model to compute the distances between the generated distribution, the Gaussian-smoothed empirical distribution, and the population distribution. Numerical simulations show that the transition to generalization occurs when the number of training samples $n$ becomes proportional to the data dimensionality $d$.

**Strengths:**

The paper addresses an important and timely problem: understanding when and how diffusion models transition from generalizing to memorizing their training data. This issue has substantial theoretical and practical implications. Understanding these regimes is crucial for real-world applications, particularly in ensuring that generative models do not inadvertently memorize and reproduce training data, which can lead to issues such as privacy violations.

**Weaknesses:**

As acknowledged by the authors themselves, this work is still in "draft" form (L410). In particular, it falls significantly short of publication standards, especially for ICLR. Below, I outline several critical issues.

1. **Circular argument:** The central argument of the paper is circular. It defines generalization as the scenario in which the generated distribution is closer to the population distribution than the (smoothed) empirical one and then claims that the transition occurs when generalization arises.

2. **Questionable modeling choices:** The choice to model the empirical distribution of the training set as a Gaussian smoothing of the actual empirical distribution is motivated solely by the divergence of the KL metric when using the empirical measure. Metrics like the Wasserstein distance are well-suited for comparisons with empirical measures. Furthermore, the chosen kernel variance (linked to the mixing time of Gaussian distributed points) appears to be unjustified and inadequate. Paradoxically, this choice minimizes the distance to the population distribution, contradicting the goal of accurately modeling the empirical distribution.

3. **Trivial statement:** Section 2.3 essentially states that the distribution of points at late times is Gaussian, which is an expected outcome by construction. This does not warrant in-depth analysis.

4. **Linear model limitations:** The paper uses a linear diffusion model arguing that a linear score function is justified because the score of a Gaussian is linear and at late time the distribution is Gaussian. Such a model can only generate Gaussian distributions and trivially requires $d$ points (in $d$ dimensions) to match the population score (see also point 6 below). This score model is strongly constrained in its expressive power and provably cannot fit the training distribution (nor its Gaussian smoothing), whose score is a weighted Gaussian convolution of the score of the single training points. Thus, **this model cannot properly memorize the training points**, contrary to the paper's claims (note that this is just due to an inadequate definition of memorization which does not require the generated distribution to be close to the training one). In fact, Biroli et al. (2024) explicitly link memorization to short times, where the score approximation of the authors is off.

5. **Inconsistent results:** Indeed, when testing on non-linear diffusion models (see Fig. 4), the results are inconsistent. Notably, the supposed memorization-to-generalization transition is not supported by the authors' own metrics, as the probability that the generated distribution is closer to the population distribution remains high ($>0.8$), regardless of the number of training samples.

6. **Inconsistent limits:** In the proportional regime, i.e., $n \simeq d$, or when $n \ll d$ – that the authors discuss from Sec. 3 – the results in Sec. 2 do not hold, as they require $n$ to grow exponentially with $d$.

7. **Practical application oversight:** The final comment of the paper on potential practical application is also problematic and too naive, as it does not take into account the quality of generations, which is crucial in practical scenarios.

8. **Neglect of the number of parameters:** The study fails to address how the degree of overparameterization influences memorization behavior, despite its known importance (e.g., Yoon et al., 2023). Instead, the focus on an overly simplistic linear model, which inherently lacks the capacity to memorize, overlooks this essential aspect.

9. **Superfluous equations:** The paper contains a significant amount of math (see, e.g., Eq. (21)) which is not well explained or linked to the empirical findings.

10. **Figures:** The figures lack essential labels and color codes, making them difficult to interpret. Error bars or standard deviations are missing, and the use of linear scales (instead of log scales) is inadequate for illustrating trends. The Hellinger distance in Figure 2(b) decreases by only 0.02, which is insufficiently significant to justify the conclusions.

11. **Writing quality:** The writing is weak, with numerous grammatical errors and poorly constructed sentences.

**Questions:**

None.

---

> ### Author Response · Authors · 2024-11-25
>
> We thank the reviewer for their feedback and will respond to the raised questions below.
>
> **1. Circular argument:**
> Thanks for this comment. In the paper we use $E_{TG}, E_{OG}$ to denote the distance between train and generated distribution and original and generated distribution respectively. Usually, by generalization, it is meant that $E_{OG}$ is small. However, according to our definition, memorization to generalization transition takes place when $P(\Delta=E_{TG}-E_{OG}>0)$ changes sharply from a smaller value to close to being one (please note the difference in definition). Even if $P(\Delta=E_{TG}-E_{OG}>0)$ is close to one, it is one condition on two variables $E_{TG}, E_{OG}$, hence there are an infinite number of possibilities for individual behavior of $E_{TG}$ and $E_{OG}$ on the onset of the transition. Please see figure 1. We have explicitly drawn some of the possibilities that might be present even if  $P(\Delta=E_{TG}-E_{OG}>0)$ increases sharply with an increase in train dataset size. One possibility is that $E_{TG}$ increases while $E_{OG}$ stays constant as the size of the training dataset is increased featuring a sharp transition in $P(\Delta>0)$. For the realistic diffusion model on the Gaussian dataset, we find that this is not the case. See figure 4 for more details on this.
>
> **2. Questionable modeling choices:**
> We thank the reviewer for this suggestion.  Our prescription for the choice of the Gaussian kernel allows us to compare a small number of generated samples to a larger training dataset by carefully adjusting the kernel variance depending on the number of points in a principled way. And our distance measure shows the important feature of memorization to generalization transition. Evaluation of Wasserstein distance between the training dataset and the generated dataset might be computationally challenging, but it is an interesting open question. Also, Wasserstein distance does not provide a smooth approximation to the probability densities it is comparing, making it a less desirable candidate for visualization purposes.
>
> **3. Trivial statement:**
> We agree with the reviewer on his/her point. Also, we would kindly like to note that the timescale $t_R$ defined near equation (14) is a property of the underlying distribution and the particular stochastic differential equation.
>
> **4. Linear model limitations:**
> In the paper, we have restricted our analysis for the linear diffusion model to isotropic Gaussian distribution. When the generated distribution is closer to the training dataset compared to the original distribution we say the model is memorizing the train data. For this specific dataset, the model is expressive enough to memorize as indicated by figure 3. We see the memorization behavior for smaller training dataset sizes. We agree that for more general probability distributions we need to modify our model. A possible idea for the modification in terms of kernels is outlined in section 5.
>
> **5. Inconsistent results:**
> We thank the reviewer for the comment. Our metric supports the transition. We would like to emphasize that the metric used for figure (4) is given in equations (23), and (24). This metric is similar to the one in equations (10), and (19) that is used for the linear diffusion model, however, they are not exactly the same. The simpler metric in equations (23), and (24) is used to make the computation faster in the realistic diffusion model. As a result of this change in metric, a slightly different numerical value for $P(\Delta>0)$ is visible in figure (4).
>
> **6. Inconsistent limits:**
> We have clearly defined the smoothened probability distributions in section 2.  The definitions remain valid for all values of $n,d$. The definitions are motivated by the analysis in the regime where $n$ grows exponentially with $d$.
>
> **7. Practical application oversight:**
> How the quality of the generated images depends on the ability of the model to generalize is an interesting question. Such questions are left to future work on this topic.
>
> **8. Neglect of the number of parameters:**
> In figure 4 we have presented results for a realistic non-linear diffusion model of 12.9  million parameters trained on an isotropic Gaussian dataset showing results similar to the linear diffusion model discussed before.
>
> **10. Figures:**
> We thank the reviewer for carefully reading our draft. We have error bars included in the plots, they are very small and not visible clearly. Figure 2(b) only makes the point that there is indeed a change of slope in the Hellinger distance squared $H^2$ between generated and original distribution on the onset of memorization to generalization transition. To see the change more clearly one can consider the plot of $(1-H^2)^{1/d}$ vs $n/d$.

---

### Note · Authors · 2024-11-25

**Comment:**

We thank all the reviewers for their feedback.

**Withdrawal Confirmation:**

I have read and agree with the venue's withdrawal policy on behalf of myself and my co-authors.